# *La Commedia Scientifica*: Dante and the Scientific Virtues

Anthea R. Lacchia[1,2] and Stephen Webster[2]

[1] Irish Centre for Research in Applied Geosciences, School of Earth Sciences, University College Dublin, Dublin, Ireland

[2] Science Communication Unit, Imperial College London, London, UK.

*Correspondence to*: Anthea R. Lacchia (lacchiaa@tcd.ie)

**Abstract.** The ethical challenges facing contemporary science range from scientific misconduct, to the rightful treatment of people, animals and the environment. In this work, we explore the role of virtue ethics, which concern the character of a person, in contemporary science. Through interviews with thirteen scientists, eight of whom are geoscientists, we identify six

virtues in science (honesty, humility, philia, innocence, generosity and reticence), paired with vices, and construct a narrative argument around them. Specifically, we employ the narrative structure of the late medieval poem *Divine Comedy*, by Dante Alighieri, and draw on its moral universe to explore the scientific virtues. Using this narrative device, we make the case for virtue ethics being a reliable guide for all matters scientific. As such, this work lays out a modern code of conduct for science.

## 1 Introduction

Modern-day science is suffering. Everywhere we turn, signs of malaise are rife. Scientific misconduct, including plagiarism, falsification and fabrication of data, has long been recognized in science (Resnik, 1998). More recently, mounting financial pressures (Ledford et al., 2019) and a culture of fierce competitiveness (Abbott, 2016; Lawrence, 2002), where mental illness is widespread (Evans et al., 2018), have been associated with the cutting of ethical corners (Casadevall, 2019) and with

scientific misconduct, such as the recent reports of genome-edited babies in China (Cyranoski, 2019).

Has science lost its way? Today's scientists, from hastily completing grant applications to feverishly editing manuscripts, are busy staying afloat in a terrifying world. As they battle the incoming waves of funding cuts (e.g. Nature, 2019a) and administrative burdens (James, 2011; Warren, 2018), ethical considerations seem few and far between.

So where should we go to find answers to modern ethical dilemmas in science? We will argue that, by turning to the writing of a late medieval Italian poet and great thinker on ethics — Dante Alighieri (1265-1321) — we will retrieve the moral compass we seek. Specifically, we will seek inspiration from Dante's moral universe, masterfully articulated in his poem *Divine Comedy* (Dante, 1922), as a means of exploring the role of virtue ethics in science.

Virtue ethics, a branch of ethics first articulated by Aristotle (384–322BC) in his *Nicomachean Ethics* (1953), henceforth referred to as *Ethics*, concern the character of a person, as opposed to his or her actions (Rachels, 2010). According to Aristotle, the 'good of man' is 'an activity of the soul in conformity with virtue' (Rachels, 2010). Further, expressed in contemporary terms, Aristotle's ethics helps us make the link between the good character of a person, and the moral integrity of that person's society, in particular the institution for which they work. For almost by definition, a virtuous person combines a sense of personal integrity with a constant search for professional diligence, as a farmer, a shop keeper, a lawyer or of course a scientist. It is a key Aristotelian point, and an alluring one, that the person who carries out their work virtuously will flourish, will have an enduring sense of well-being, and will contribute to the proper development of their community. While Aristotle had in mind the city state of Athens, and its citizens, this paper argues that the same ideas can be made to apply to science, and its scientists. In particular then, if we wish to be assured that science has its house in order, morally speaking, we must look first to the life of the individual scientist, and its proper development, rather than rely solely on impact measurements, codes of conduct and policy reforms. Of course, the institution is important — we can hardly imagine the scientific profession without them. But the Aristotelian point is that first of all we must look at the scientific self. The starting principle is that, in order to be 'good', one must cultivate the virtues, and embody them. The task is far from trivial. The responsible scientist is the one who establishes what these virtues might be, and makes a habit of them, whatever

the difficulties presented by the prevailing culture. It is towards that end, the elaboration of the scientific virtues, that the arguments of this paper are organised.

Aristotle would counsel that the effort is worthwhile. For, by diligently following the precepts of good practice, and by seeing the cultural benefits of establishing fair dealings with others, a scientist will be establishing within themselves a set of 'internal goods'. From that, the Artistotelian would argue, benefits will ripple out to the wider professional culture – to science. Moreover – a key Aristotelian point – the work will bring pleasure: the scientist, secure in the knowledge that they are working well, according to the best traditions of their craft, will have the sense that they are flourishing. The important question that follows is this: in contemporary science, is the 'good' scientist also likely to be a successful one? Conversely, might it be that 'goodness' in science is a professional hazard in today's research culture? Does the nature of research culture act against the possibility of a scientist developing their personal and professional virtues? It is the challenging and disturbing nature of these questions that make Aristotle's 2,400-year-old teachings so pertinent to the modern-day scientist.

Philosopher Elizabeth Anscombe (1919-2001), in her seminal paper *Modern Moral Philosophy* (1958), restored virtue ethics to a position of prominence in the contemporary ethics debate (Rachels, 2010). Since her work, some applications of virtue ethics to science have surfaced (MacFarlane, 2009; Resnik, 2012). However, as Anscombe argued (1958), there are 'great contrasts' between modern moral philosophy and Aristotle's ethics. Indeed, since the advent of Christianity, ethicists have focused on actions rather than character, and the virtues have been cast aside (Rachels, 2010). Scientists are still less inclined to consider ethics beyond the confines of consent forms and issues over data protection (see Methodology for details about quotes):

*"We think ethics doesn't concern us, because ethics has to do with doing unpleasant things to animals or people and we don't do that; so, we learn to very, very quickly tick the 'no' box on ethics [forms]. Nobody ever asks you about the virtues of a good scientist."*      - Professor

Hence, for many contemporary philosophers and scientists, gazing at the virtues will be like gazing directly into the sun. The issues are so strong, so burningly important, that paradoxically they are hard to apprehend clearly. Before going any further, therefore, we must equip ourselves with a narrative device to let us see them properly, in spite of their brightness.

Just as the mythical hero Perseus, in order to avoid being turned into stone by the gaze of the monster Medusa, held up a reflecting shield to take aim and cut off her head (Palazzi, 1988), we will deploy a narrative device to gaze at the scientific virtues. Specifically, we will seek inspiration from Dante's *Divine Comedy*, *Divina Commedia* in Italian (from now on referred to as *Commedia*), a literary masterpiece that has inspired almost 700 years of research and scholarship (Belliotti, 2014).


In the *Commedia*, Dante imagines making a metaphorical journey through Hell (*Inferno* in Italian), Purgatory (*Purgatorio*) and Paradise (*Paradiso*). Each of the three stages is written as a section (*cantica*) composed of 33 parts (*canti*), with one additional introductory *canto* (Reynolds, 2006). The *Commedia* starts from a moment of 'profound psychological crisis' and tells the story of how it is resolved (Shaw, 2014):


> Nel mezzo del cammin di nostra vita
>
> mi ritrovai per una selva oscura,
>
> che' la diritta via era smarrita.
>
> (*Inf*. I, 1-3)


> (In the middle of the journey of our lives
>
> I found myself in a dark wood,
>
> for the straight path was lost.)

Dante, who sets his fictional journey through the underworld in the thirty-fifth year of his life (Scott, 2004), invites the reader to identify with what today we would call a 'mid-life crisis' (Shaw, 2014). In this sense, the *Commedia* can be seen as

a precursor to the modern self-help genre (Shaw, 2014). Indeed, any scientist who has faced an ethical dilemma could relate to Dante's verse.

O' voi, ch'avete l'intelletti sani,

mirate la dottrina che s'asconde

sotto il velame de li versi strani.

(*Inf.* IX, 61-63, Reynolds)

(O ye whose intellects are sane and sound,

note well the doctrine that beneath the veil

of the mysterious verses can be found.)

With these lines from the *Inferno*, Dante exhorts his readers to pay attention to the allegorical meaning hidden behind his verse (Reynolds, 2006). For him, poetry is a vehicle of truth, and his poetic verses, if properly interpreted, become a pedagogical device that allows the reader to contemplate truth (Auerbach, 1971). The poem's didactic and campaigning intent is manifest in several passages of the *Commedia* (Honess, 1997), such as in *Paradiso*, when Cacciaguida, Dante's ancestor, exhorts Dante to share the vision of divine justice and truth he has been privy to, when he returns back to Earth:

Ma nondimen, rimossa ogni menzogna,

tutta la tua vision fa manifesta;

(*Par*. XVII, 127-128)

(But nonetheless, with all lies removed,

all your vision make manifest;)

Clearly, Dante's mission can be thought of as a communicative one (Honess, 1997; Shaw, 2014). We will draw upon the *Commedia*'s broad narrative structure, as well as its communicative intent, as we embark upon a journey of our own, in order to explore modern ethical dilemmas in science and the role of virtue ethics in solving them. As we shall see, this narrative device will allow us to contemplate questions of ethics in a more personal, direct and practical way than do the myriad rules and principles that fill textbooks on science ethics.

Unlike Dante's divinely willed journey, ours will be secular. Naturally, in omitting the religious aspect, we omit an important part of the *Commedia*'s exegesis, but not, we trust, the essence of this masterpiece as a description of a moral universe. In choosing narrative as a form of argumentation (Calame, 2005, p.66), we follow the example of the Greek poet Sappho (Voigt, 1971), the philosopher Plato (1974) and, more recently, contemporary author Jostein Gaarder (1991), as well as Dante.

The *Commedia* is full of references to Dante himself and to the characters of contemporary Florence, as well as to historical, mythological and biblical figures. Dante is accompanied on his journey first by the great Roman poet Virgil, and then by Beatrice, traditionally identified with Beatrice Portinari, the woman Dante loved in his youth (Shaw, 2014). In this work, we will encounter characters from ancient Greek and Roman myth and philosophy, such as Aristotle, who will help us make our journey through the *Inferno*, *Purgatorio* and *Paradiso* of science, and we will exercise literary freedom in projecting onto them modern scientific concerns. For now, suffice it to say that we will encounter three main characters: a Scientist, and two guides, first Ulysses and later Aristotle. In our choice of characters, we weave together the thoughts and the questions of the Scientist, Ulysses and Aristotle, and Dante the character, whose journey we follow, Virgil, who represents reason, and, finally, Beatrice, who represents divine knowledge. But before we set off, let us consider Dante's world and how it can speak to modern science.

Dante's work cannot be understood without understanding the political turmoil of Florence in the poet's time. Indeed, Dante's complicated relationship with his native Florence permeates the *Commedia* (Hainsworth & Robey, 2015). The Florentines and their feuds are particularly well represented in the *Inferno*, as these ironic verses attest to:

Godi, Fiorenza, poi che se' sì grande

che per mare e per terra batti l'ali,

e per lo 'nferno tuo nome si spande!

(*Inf.* XXVI, 1-3)

(Rejoice, Florence, for you are so great,

that over sea and land you beat your wings;

and through Hell your name echoes!)

In Dante's lifetime, Florence was one of many city-states in northern and central Italy (Scott, 2004). A wealthy mercantile and banking centre, it was deeply troubled by factional struggles and changes in legislation and systems of government (Shaw, 2014). At the turn of the fourteenth century, Florence was a city divided, with nobles and wealthy merchants splitting

into Guelfs and Ghibellines, respectively the party of the Pope, and that of the Holy Roman Emperor, although petty rivalries tended to override ideological affiliations (Shaw, 2014). The inhabitants of Florence did not just have internal feuds to contend with, but also spats with other Italian cities. As pointed out by writer Andrew Norman Wilson in his book *Dante in Love* (2011, p.39):

'The inhabitants of medieval Italian cities lived in a state of such enmity with one another that it was necessary for them to live huddled in fortified towers.'

The Ghibellines came to power in Florence in 1260, following the battle of Montaperti, but were in turn expelled from the city in 1266, after the Guelf victory at the Battle of Benevento (Santagata, 2016; Shaw 2014). In 1295, the Guelf party in

Florence divided into two factions, known as Blacks and Whites (Santagata, 2016), a split that would lead to Dante's traumatic exile. Dante was an ambitious man, actively involved in public life. By 1300, Dante's party, the Guelfs, had been in power for more than thirty years (Shaw, 2014). From June to July of 1300, Dante, who aligned with the White Guelfs, served as one of six priors who exercised executive power in Florence. In 1301, while he was absent from the city, the Blacks staged a coup and returned to power, partly owing to an intervention by Pope Boniface VIII (Wilson, 2011). The following year, Dante was fined, charged with corruption, and exiled. It is in exile that he started work on the *Commedia* (around 1308), a task that would consume him until shortly before his death (Shaw, 2014).

An interesting parallel emerges between the disordered state of late Medieval Florence and the state of science today. Let us consider Shaw's (2014, p.37) eloquent description of Florence in Dante's time:

'Florence, with its never-ending series of changes in legislation, political institutions, and even the makeup of the body politics, is like a sick woman tossing and turning on her bed, whose constant restless motion seeks to assuage her pain but is itself an expression of her malaise. The struggle to institute, fine-tune and protect the fledgling institutions of democratic accountability and the rule of law against the vested interests of the *grandi* [aristocrats] would seem to be incompatible with peaceful and healthy civic flourishing.'

Considering the repeated calls for new regulations in genetic research (Nature, 2019b), the everlasting quest for new ways of measuring scientific success (Curry, 2018), and the ceaseless drafting of new codes of conduct for researchers (Hiney, 2018), science resembles that sick woman, forever tossing and turning through restless nights.

Dante recognised that his native Florence was in dire need of moral guidance and sought to save her through his writings. Given the ethical issues that permeate science today, it is clear that we need a Dante for science. This is precisely the mission that provides impetus to this work.

## 2 Methodology

In this study we have sought to apply notions of virtue ethics to science. We use the term science to denote the natural, physical, and social sciences, but we wish to point out that many of the virtues and vices identified in this paper are pertinent to academic research and issues of research integrity as a whole. Our choice to restrict our investigation to science which, because of its high research costs and societal impacts, is in truth a zone of particular academic pressure, is reflected in our choice of interviewees (see below).


In order to identify those virtues most pertinent to modern-day science, we carried out 13 one-to-one interviews with scientists from various disciplines and institutions, and at different career stages. Our sample included: two emeritus professors, five professors (including principal investigators, henceforth referred to as PIs), two postdoctoral researchers (postdocs), two PhD students, one technical officer and one policy officer. Participants were affiliated with six different institutions in Ireland, the UK and Belgium. We recruited participants drawing on a convenience sample of academics,

including collaborators, colleagues, former supervisors and researchers whose views on research ethics we were aware of (through academic papers, news articles or social media). The expertise of interviewees included: biology, geoscience, palaeontology, social science, immunology and medicine. Despite a relatively small sample size, our sample included a range of ages and career stages, and a gender ratio of 7 females to 6 males, although we did not formally gather demographic

information.

The interviews were semi-structured and most of them (ten) were conducted face-to-face, with three taking place via Skype or telephone. Their duration was 20 minutes to just over one hour. Prior to the interviews, interviewees were provided with an information sheet and consent form, which are included in Appendix 1. All interviewees signed the consent form and

indicated whether they wished to remain anonymous. Interviews were recorded electronically and transcribed.

The purpose of the interviews was two-fold: firstly, by means of thematic analysis (Boyatzis, 1998), we identified a series of virtues described by the scientists during the interviews. These virtues acted as a framework on which we constructed the Dante-inspired narrative, with each virtue paired with a sin (see Appendix 2 for detailed structure of the narrative). Secondly,

the interviews served as a source of quotes for our narrative. Overall, the framework of virtues and the quotes — which are presented in italics throughout the text alongside the interviewee's title or career stage — serve to ground our arguments in reality. Further quotes in the narrative are derived from the works of the characters themselves, such as Aristotle, who will read from his *Ethics* (1953). But let us embark upon our journey without further delay.

## 3 A dark wood

Dante set his *Commedia* in the Easter week of 1300. Our *Commedia Scientifica* starts in 2021, 700 years after Dante's death. Let us turn our attention to a lonely figure sitting at a lab bench: from now on, we shall refer to her as the Scientist. Her biography on the departmental website tells us she is completing her third postdoc. Under her name, we see an impressive list of awards and publications. What the Scientist's biography does not mention is that her parents are worried about her. She has not been eating properly lately. Her grant has nearly run out. Last week, her lab manager asked her to 'amend' a figure and a table. Just a small tweak for the sake of statistical significance. She knows she needs to get the results they need, and she needs to get them fast. She is *so* close to getting that lectureship. Changing a number or two is easy to do, she tells herself. Yet, she cannot bring herself to do it. Exhausted and not knowing how to proceed, she rests her head in her arms. A document entitled *Singapore Statement on Research Integrity* (2010) lies open beside her. Without noticing, she falls into a deep sleep. Let us trace her footsteps as she dreams.

The Scientist awakes in a dark wood. Moonlight cuts through a dense canopy of trees, merging with mist rising up through the humid air. Distant howling punctuates the silence. Terrified, she rises and makes her way through the forest, eventually reaching a clearing. Through the mist, she can just about make out the outline of an imposing building perched on a hilltop, complete with turrets and barred windows.

She hurries towards it. A tall, bearded man is standing outside the building walls, a small dog at his feet. He waves to her, and seems to beckon her to come closer. 'Can you tell me where I am, please?' she asks. 'Of course,' he replies. 'I have been

waiting for you. You have reached the place where lost scientists are forever lost. Once, I too was one of these lost souls. But I have learned from my mistakes. Ulysses is my name and I will guide you through these unhappy walls.'

This self-proclaimed guide is none other than the heroic wanderer Ulysses (Homer's Odysseus), the king of Ithaca who took part in the ten-year siege of Troy described in Homer's *Iliad* (Homer, 2014), and whose ten-year return journey to Ithaca is the subject of Homer's *Odyssey* (Homer, 1996). At his side sits the faithful dog Argos (Boyde, 2000).

Dante, having no knowledge of Greek, learned about Ulysses from Roman scholars such as Ovid, Cicero, Seneca and Horace
(Bosco & Reggio, 2010; Boyde, 2000). In the *Commedia*, Dante places Ulysses among Hell's fraudulent counsellors (*Inf.* XXVI), and it is here that the hero delivers his *orazion picciola* ('little speech', *Inf.* XXVI, 122) to his crew, persuading them to sail with him into the unknown:

Considerate la vostra semenza:
255             fatti non foste per viver come bruti,
             ma per seguir virtute e canoscenza.
             (*Inf.* XXVI, 118-120, Singleton)

(Consider your origin:
260             you were not made to live as brutes,
             but to pursue virtue and knowledge.)

In Dante as in Homer, Ulysses is an intensely curious man, with an unquenchable thirst to know more about the world, as well as being an accomplished orator, well versed in the art of persuasion (Bosco & Reggio, 2010). In Ulysses' speech, the
two values that define what it is to live as human beings — intellectual activity (*canoscenza*) and practice of the moral virtues (*virtute*) — conform to what Aristotle describes as the source of human happiness (Shaw, 2014, p.129). This is not

the only time Dante draws from Aristotle on matters of ethics. Furthermore, as we shall see, these values are also entirely relevant to science ethics.

In Homer, Odysseus eventually reaches the shores of his beloved Ithaca and reunites with his wife Penelope (Homer, 1996). But Dante's Ulysses meets a very different fate. Having dared to venture beyond the Pillars of Hercules (which we recognise as the Straits of Gibraltar), the hero goes beyond what is allowed to human beings (Chiavacci Leonardi, 1991; Poirier, 2016; Reynolds, 2006; Shaw, 2014). This transgression, which Dante calls *folle volo* ('mad flight', *Inf.* XXVI, 125), is punished with death, as the ship is swallowed by the sea (Dante, 1922). This punishment makes sense in the context of Dante's moral

universe, where human reason cannot reach truth, if unaided by divine knowledge (Bosco & Reggio, 2010). Ulysses may be punished, but he is also admired by Dante, who recognises in him much of his own passion for knowledge and fierce ambition (*La sete natural che mai non sazia*, 'The natural thirst which is never quenched', *Purg.* XXI, 1). Indeed, there is a lot in common between Dante's own divinely willed 'flight' into the underworld and Ulysses' *folle volo* (Bosco & Reggio, 2010).


In Ovid's *Heroides*, Penelope urges Ulysses to return to Ithaca, reminding him of his duties as a father, son and husband (Boyde, 2000). Her pleas are summarised by Dante's Ulysses in these lines:

> Ne' dolcezza di figlio, ne' la pieta
>
> del vecchio padre, ne l'debito amore
>
> lo qual dovea Penelope' far lieta,
>
> vincer potero dentro a me l'ardore
>
> ch'i ebbi a divenir del mondo esperto
>
> e de li vizi umani e del valore;
>
> (*Inf*. XXVI, 94-99)


(Neither fondness for my son, nor reverence

for my aged father, nor the rightful affection

which would have made Penelope glad,

could conquer inside me the longing

295          that I had to gain experience of the world,

and of human vice and virtue.)

These wonderful verses describe Ulysses' urgent longing to gain knowledge and satisfy his curiosity — a force inside him so strong that it leads him to shirk his familial duties. Thus, Dante's Ulysses reminds us of the danger of taking knowledge —

and science — too far, as already recognised by Dantean scholars Patrick Boyde (2000), Jean-Louis Poirier (2016) and Prue Shaw (2014, p.130). Clearly, Ulysses is somewhat of a mixed character: vengeful, boastful, neglectful, as well as cunning and brave. This ambivalence is also a trait that has long been identified as pertaining to scientists by sociological studies of scientists (Mitroff, 1974). Indeed, by Mitroff's account, the ability to hold diametrically opposing views in mind at the same time may be essential to science.


In Homer, as pointed out by contemporary philosopher Alasdair MacIntyre (1984, p.132), Odysseus' cunning is treated as a virtue, 'and it is of course for his exercise of the virtues that a hero receives honour.' We too, without forgetting Ulysses' darker side so well epitomised in the verses above (*Inf.* XXVI, 94-99), will take the kinder view, as we follow Ulysses through the scientific underworld. Just as, in Dante, trusted guide Virgil personifies human reason and knowledge (Shaw,

2014), so we will take our Scientist's guide, Ulysses, to personify the intellectual curiosity and love of knowledge that scientists recognise in themselves:

*"It's a bit like being a mountain climber, a polar explorer or a deep-sea diver. You go and explore somewhere, you go*

*somewhere nobody has been before. […] You are exploring uncharted territory. There is that tremendous excitement that*

*you get."*                                                                                                                                                  - Professor

Given the spirit of science which, irrespective of author, Ulysses embodies, it is fitting that we should seek guidance from him on where to place our own Herculean Columns of the life scientific. But let us return to the Scientist and her dream. Ulysses strokes Argos and whispers that he will be back soon. He starts to walk towards a large wooden door. The Scientist
follows, transfixed...

## 4 *Inferno* - the sins of a science that has lost its way

Lasciate ogne speranza, voi ch'intrate.

(*Inf.* III, 9)

(Abandon all hope, ye who enter here.)


This inscription is scrawled above the door of our unethical research centre, a fiery pit of misconduct and infinite sadness. Before proceeding, it is worth noting that the sins we will encounter here and in our scientific *Purgatorio* will only broadly follow those of Dante's moral universe. In common with the *Inferno*, we will consider incontinence, violence and fraud (Auerbach, 1961; Di Zenzo, 1965), but substantial deviations from the *Commedia*'s nine circles and seven terraces of
purgatory must be allowed, if pertinence to contemporary science is to be our guiding principle.

The Scientist and Ulysses step inside a vast, open-plan office, complete with rusty scientific instruments and dusty lab benches. In one corner, dozens of scientists are seated at long rows of desks, quietly sobbing. 'These,' says Ulysses, 'are the souls of scientists whose plight I know all too well, for their curiosity went too far.' Heavy chains hold the scientists down,
forcing them to stare forever into microscopes with broken lenses, resembling the instruments they held so dear in life. In Dante, punishments must always reflect the nature and gravity of the sin — a principle he calls *contrappasso* (Scott, 2004) — becoming 'either an externalisation of the aberrant impulse or a corrective to it' (Shaw, 2014, p.113). The same is true of the unethical research centre we are intent on exploring.

As they make their way across the dimly-lit office, Ulysses and the Scientist see another group of souls huddled together. As they approach them, they are met with loud, desolate cries: the scientists are chained to their desks, but are looking up, arms outstretched, towards towering piles of books and documents. 'These souls are separated from their families by walls of textbooks and papers. They can hear their loved ones calling for them from behind these walls, but, alas, they are never to be reunited,' says Ulysses. Turning away from the tormented souls, he wipes away a tear from his cheek and walks on.


In the centre of the office, hundreds of souls are crawling on their hands and knees, weighed down by heavy plaques hanging from their necks. Some plaques are decorated with inscriptions such as 'Dr', 'Professor' and 'Emeritus', followed by the names of universities and funding bodies; others simply bear a series of figures on them. 'These,' says Ulysses, 'are the souls of scientists who were greedy for money, power or fame (e.g. Chawla, 2019). The heavy plaques they are dragging in
endless circles on the ground bear the insignia of the science they hungered for in life. Some of the plaques show the amount of money they won in grants, or the titles and affiliations they treasured above all else, while others bear the words of modern scientists:

*"You have senior scientists making all the decisions for junior scientists. Those senior scientists often don't have to live*
*with the consequence of some of the decisions that they're making. I think that's a big issue to tackle, to rebalance*
*power. [...] People get seduced by the power."*                                                                                                    - Professor

*"[If] you're in a prime position to get money for funding, that automatically, I'm not saying everybody, but it inflates*
*egos. [...] It definitely establishes a power dynamic of having to chase the big guy with the money, and the smaller people*
*have to scramble around - that is unhealthy competition."*                                                                      - PhD student

*"One day a week everyone would go down for tea and coffee and cakes and, you know, just one between the group and everything. My supervisor said: 'That's a waste of time. In that one hour, you could see two more patients and generate more data."*

- PhD student

At this point, the Scientist's gaze is drawn to another side of the room, where scientists are chained to one another and forced to share tiny desks. They elbow each other angrily, as they try to claim some desk space as their own. 'Tell me, dear guide,
what sin plagues these poor souls?' asks the Scientist. 'These scientists, once PIs, fostered an atmosphere of fear, secrecy and competition in their workplaces,' explains Ulysses. 'Now, they are forced to co-exist in this confined space, while the luxurious offices they occupied in life lie empty behind them,' he says.

The Scientist shudders at the sight of one of the souls who, hunched over, is desperately trying to hide a paper from the
glances of his neighbours. It is worth noting at this point that, in Dante's underworld, the damned feel no real regret for their sins, only sadness for what they have lost (Sermonti, 2018). The same is true in our sad, scientific underworld, where the Scientist is peering at the words of modern scientists, etched on the walls:

*"The scientific career is super competitive. Of course, to get all those publications you need to be super competitive. I
don't like this face of science; I don't like how people just change their character."*

- Postdoc

*"So many times I've been told 'don't work with that person', and not been told why."*

- PhD student


*"I've had a tough period of three years. I just want to finish and leave this toxic environment."*

- PhD student

*"I sent this co-author the draft [of my paper] and he sent it back to me with a sarcastic comment about how I'd left this other person off, [saying] that it was obviously a mistake, even though she was clearly in my acknowledgements at the end. I had acknowledged her because, to me, she had done a level of work that only warranted an acknowledgement. [...] But then we had a group meeting [...] and he [my supervisor] said we should just put this person on all the papers because 'if I want to work with the lab it'll just be easier if we do this.' [...] That's not right, to give someone token credit just because it makes it easier than not doing it."*
                                                                                                                        - PhD student

At this point, the atmosphere of mistrust and competition becomes so overpowering that the Scientist feels weak and has to hold on to Ulysses. Together, they stumble through a small door and an old laboratory comes into view, illuminated by red, flickering lights. Sounds of growling pierce the thick air. To her horror, the Scientist sees dozens of animals squeezed inside tiny cages. Separate groups of animals are held in pens, while souls of scientists clean their empty cages.

'What dreadful place is this?' asks the Scientist. 'Here, you see the souls of those scientists who were violent in life,' says Ulysses. 'Those who are guilty of abusing the power they held over people in their charge are forced to do the menial tasks they avoided in life.' The Scientist turns pale, as she remembers the words of junior colleagues who suffered power abuse at the hands of their supervisors:

*"That's something that worries me, that once I leave and we have publications coming out of the study, for which I have done every single patient visit, collected every bit of data, I might not get first author. I might not even get co-authorship."*
                                                                                                                        - PhD student

'Those who were cruel towards non-human animals in the name of science are imprisoned in these cages, transformed into the sentient beings they tormented in life,' says Ulysses, before silently pointing to an engraving on the wall behind the cages:

Our throats were dry and thirsty; we drank deep;

And then the demon goddess lightly laid

415                  Her wand upon our hair, and instantly

Bristles (the shame of it! but I will tell)

Began to sprout; I could no longer speak;

My words were grunts, I grovelled to the ground.

I felt my nose change to a tough white snout,

420                  My neck thicken and bulge. My hands that held

The bowl just now made footprints on the floor.

(*Met*. XIV, 277-284, Melville)

With these lines, which our guide Ulysses will relate to very well, Roman poet Ovid (43BC- 17AD) described the famous

spell cast by enchantress Circe, which transformed Odysseus' crew into swine. This is just one of the many, vivid

transformations in Ovid's *Metamorphoses* (Ovid, 1986), which inspire the punishments in this division of our unethical

research centre.

'In the darkest corner of the room,' whispers Ulysses, 'a special cage is reserved for those scientists who conducted

experiments on other humans. They are forever forced to experience the pain of the experiments they themselves devised.'

Keen to leave this cruel menagerie of unethical science behind, the Scientist and Ulysses hurry through a dark passage,

which opens out into a vast lecture theatre. Endless rows of seats surround a central pit, where a fire is burning. Paintings

depicting the tragic story of Daedalus and Icarus decorate the walls. According to the Greek myth, Daedalus was an inventor

who built a labyrinth on Crete, only to be held there as a prisoner by Minos, the king of Crete (Palazzi, 1988). In order to

escape, Daedalus engineered wings of wax and bird feathers for himself and his son Icarus (Aimonetto, 1957; Palazzi, 1988).

Not heeding his father's warnings not to fly too close to the sun, Icarus soared upwards, melting the wax and falling to his death (Palazzi, 1988). His final moments are described by Dante:

ne' quando Icaro misero le reni

440         senti' spennar per la scaldata cera,

gridando il padre a lui: 'Mala via tieni!'

(*Inf*. XVII, 109-111, Singleton)

(nor when the wretched Icarus

445         felt his loins unfeathering by the melting wax,

and his father cried to him, 'You go an ill way!')

'In this room,' says Ulysses, 'dwell the souls of scientists whose sin was excessive pride.' He points to the fire in the centre of the lecture theatre: 'The scientists tending to those flames are forced to burn the manuscripts, papers and books they wrote

in life. Published or unpublished, it is all the same: the fruits of their labour are lost forever in those pyres'.

Nearby, souls tremble with anger as they stare at portable computer screens chained to their laps. 'What is causing them such rage?' asks the Scientist. 'They are looking at papers to which they put their names, as authors, in life,' says Ulysses. 'They are forced to watch as their names slowly move along the author list, gradually losing prominence, before landing in the

acknowledgements section, where, eventually, one by one, the letters vanish irrevocably,' he explains. The Scientist smiles sadly, recognising the vanity and hubris — from the ancient Greek ὕβρις, the feeling of being as grand and powerful as the gods (Van Hooft, 2006) — with which science is tainted.

At the back of the lecture theatre, souls are standing on a podium, intent on giving a lecture, but the words are coming out
twisted and garbled from their mouths. Beside them, souls gasp as they point at projections of their h-indices on large
screens. From h-index, a measure of scientists' productivity based on number of citations of their papers (Wendl, 2007), to
journal impact factor, a measure of the average number of citations that articles published in a journal have garnered over the
previous two years (Else, 2019a), the Scientist is altogether too familiar with this modern currency of scientific success,
which leads people to reduce one another to numbers (e.g. Van Noorden & Chawla, 2019). 'These souls,' says Ulysses, 'are
watching their cherished measures of success plummet over time on giant graphs, as other scientists take notes and model the
trends.'

*"It always saddens me when I see somebody putting their h-index in their email signature. [Or thinking] 'my papers are*

*only worth what the impact factor says they're worth'. That's a terrible thing to do, because there's an interesting story*

*behind every paper. [...] It's that story which, I think, is the important thing, not just the numbers."*

- Professor

At this point, Ulysses points to an opening in the wall. A spiral staircase leads them to the basement, the last division of
Infernal science. 'Here, we will encounter the corrupt souls of those scientists who were dishonest in life. They are guilty of
scientific misconduct — including falsification, fabrication and plagiarism,' says Ulysses. Through an endless array of news
headlines projected onto giant screens, these fraudulent souls are forever forced to re-live the moment their dishonesty was
exposed (see Enserink, 2017), as their colleagues point and stare. 'They hide their faces in shame and mourn their reputation,
which is forever lost,' says Ulysses.

*"In my role as peer-reviewer, I have come across tonnes of plagiarism, multiple times."*

- Technical officer

Ulysses reaches for the Scientist's hand. 'Come now, dear Scientist. We have reached the end of this sorry place.' Together they walk down a set of stone steps leading out to a pier, where Argos is patiently waiting. Stars illuminate a small boat anchored nearby.

e quindi uscimmo a riveder le stelle.

(*Inf.* XXXIV, 139)

(and thence we came forth to see the stars once more.)

## 5 *Purgatorio* - striving for a better science

The purgatory of science ethics is littered with the souls of those who are striving to make science better. The souls who inhabit this realm are present-day scientists, conjured up by the Scientist in her dream. Just as Dante included in Purgatory those souls whose love of God was misdirected (Auerbach, 1961; Dante, 1922; Shaw, 2014), so our Scientist will encounter here those whose love of science is misdirected.

amor sementa in voi d'ogne virtute

e d'ogne operazion che merta pene.

(*Purg.* XVII, 104-5)

(love, the seed in you of all virtue

and of every action that deserves punishment.)

Past the breaking waves, Ulysses and the Scientist reach a sandy shoreline, from where a tall, glass building can be seen. An old man is perched on a rock. He holds a dusty book in his hands and appears to be deep in thought. As they walk towards him, the Scientist makes out the title of the book: *The Nicomachean Ethics*. This lone figure is none other than Aristotle, the Greek thinker whose towering influence on ethics persists up to the present day.

Aristotle's philosophy encompassed all branches of knowledge, including natural science (Drake, 1980). His scientific results, fruit of much observation and gathering of evidence, remained unrivalled for two millennia (Barnes & Kenny, 2014), through Dante's lifetime, earning him the title of first scientist (Drake, 1980; Hankinson, 1995).

Dante placed Aristotle among the *spiriti magni*, or virtuous pagans, in Limbo, where he refers to him as *'L maestro di color*
*che sanno* ('The master of those who know', *Inf*. IV, 131). Moreover, in Dante's *Inferno*, the general classification of evil deeds is inspired by Aristotelian ethics (*Nic*. VII), which Virgil calls *la tua 'Etica'*, 'your ethics', *Inf*. XI, 80, (Scott, 2004). Aristotle's lectures on ethics, the notes of which are effectively published as his *Ethics* (MacIntyre, 1984), showcase his excellent teaching abilities, as well as his ethical theories (Thomson, 1953). We will draw on both elements as we continue our journey.


On the shore of our purgatory of science, Aristotle greets his visitors with a smile. At this point, Ulysses turns to the Scientist and says:

> e s'e' venuto in parte
>
> dov'io per me piu' oltre non discerno.
>
>          (*Purg*. XXVII, 128-129)

> (and you have reached a place
>
> From where I alone can discern no further.)

These words are spoken by Virgil in Earthly Paradise, where he leaves Dante in the capable hands of Beatrice, who is to be the pilgrim's guide from then on (Scott, 2004). While the guidance of fellow-poet Virgil was appropriate as far as the top of the mountain of *Purgatorio*, Beatrice must take over on the final stretch towards heaven. As ever, Dante's meaning is allegorical, for human reason, personified by Virgil, cannot lead to the truth without the aid of divine knowledge, personified

by Beatrice (Dante, 1922). Similarly, in our scientific journey, knowledge and curiosity alone, personified by Ulysses, cannot create good science without the aid of virtue ethics, embodied by Aristotle. Indeed, who could be more suited to guide us towards the bliss of ethical science than the pioneer scientist who is also credited with the invention of ethics (Scott, 2004)?

With a wave, Ulysses sails away into the starry night, towards two towering columns in the distance. Aristotle stands up and, quoting his *Metaphysics* (1957, p.1), says: 'All men by nature desire to know.'

'However,' he adds, 'as you yourself will have observed in the sad place you dwelled in for a time, knowledge alone is not enough to ensure the practice of good science, for it must be tempered by ethics.' With that, Aristotle directs the Scientist to the building before them. 'The souls you will encounter inside go about their business, chasing after fantasies. Unlike the realm you have left behind, no additional punishments are needed here, for science as we know it is already suffering,' says Aristotle. The Scientist nods and follows him in, through revolving doors.

A bright, modern room opens out in front of them. 'Here, reside the souls of scientists whose love of science is misdirected, for they are trying to save science by sticking like limpets to a simple view of a simple scientific method,' says Aristotle, pointing towards a group of scientists walking around in circles, reciting Merton's norms (Merton, 1973) under their breath, ad infinitum, a kind of liturgy. The Scientist recognises those utterings as the norms, or behaviours, described by American sociologist Robert Merton (1910-2003) when during WWII he sought 'institutional imperatives' that would be crucial to the ethos of science (Merton, 1973). These are: universalism, the notion that scientific claims should be evaluated irrespective of social or personal factors; communism, the idea that scientific findings belong to, and should be shared with, society; disinterestedness, the idea that scientists are not motivated by self-interest; and organized skepticism, the notion that scientists must subject their ideas to careful scrutiny. Aristotle, that great observer of how things are, cautions us however against the trap of oversimplifying human behaviour just to make us feel better: 'If only humans could be so pure,' he declares, and then: 'Nowhere in science can so clear a gaze be found!'. And the far-sighted Greek points to Ian Mitroff's

(1974) portrayal of capricious, ambitious scientists working on the Apollo lunar missions, all of them 'looking both ways'.

Mitroff, confronted by the truly complex paths of contemporary scientists, and their abandonment of the 'one way to truth', duly formulated, for each of Merton's norms, an opposing version, a liturgical heresy, the so-called 'counter-norms'. He argued that both norms and counter-norms existed in science, and had to. For instance, he writes (Mitroff, 1974, p.579), 'if universalism is rooted in the impersonal nature of science, an opposing counter-norm is rooted in the personal character of science.' He made central then to the scientific effort an unstable and oscillatory ambivalence. Naturally, for scientists

unwilling to reflect honestly on their lives, and facing hard battles for grants and esteem, the time-consuming and apparently inefficient dynamics of professional ambivalence become a source of trouble and anguish. 'These scientists desperately seek a moral compass,' says Aristotle. For each norm they so wistfully whisper, the words of the scientists interviewed by Mitroff materialise on a giant whiteboard behind them, and taunt them: 'Science is an intensely personal enterprise, it knows no simple rules,' the Scientist reads. Aristotle looks gently at our struggling hero and speaks: 'These scientists seek simple

truths, but forget the fraught ways of all professional culture. In running from that truth, they cast themselves into a place where nothing is as it seems, and where no door leads to the contentment of a flourishing career.

'Have you learned well? Yes? Then let us proceed, dear Scientist,' says Aristotle, and gestures towards a table where a group of scientists work together. The Scientist edges towards them. 'These souls are intent on devising new ways to evaluate the

success of scientists,' (Curry, 2018; Wouters et al., 2019) says Aristotle. 'In an effort to come up with metrics that will accurately measure good science, they scribble formulae and equations on large sheets of paper,' he says.

*"You can use metrics, but you must do so responsibly. But, there's a tremendous temptation for the numbers to take over, and you always have to be really mindful about how you're using them and be open about the fact that you're using*

*them. [...] Doing a piece of science is not just about good design, but bringing together the right team, establishing the right collaborations, sustaining them over the years and winning the funding ticket. The final product of the paper is the result of a very complex set of human activities, and all of those components really ought to be considered when you are evaluating something."*

\- Professor

Behind the souls, a painting of Francis Bacon (1561–1626), the English philosopher credited with the invention of the scientific method (Zagorin, 1998), hangs on the wall. 'These souls worship Bacon's scientific method, forgetting that it is the source of all misery in science,' says Aristotle.

Bacon's principal project was to renew and modernise natural philosophy, which at the time, remained heavily influenced by
Plato, Aristotle and their philosophical tradition (Zagorin, 1998). As he wrote in his 1620 work, the *Novum Organum*, people 'fall in love with particular pieces of knowledge and thoughts', and, as such, tend to 'distort and corrupt' any study they undertake 'to suit their prior fancies,' Aristotle being the prime example of this tendency (Bacon, 2000, I, LIV).

Bacon believed that the mind is plagued by so-called 'Idols', fallacies obstructing the progress of knowledge (Zagorin,
1998), and proposed a new method of discovery, based on induction, which would lead to true progress (Jardine, 2000). According to Bacon, only rigorous observations coupled with experimentation, and the formulation of scientific theories — assuming the idols are kept firmly in check — will lead to new knowledge (Zagorin, 1998).

For Bacon, the Herculean columns which decorate the title page of the first edition of the *Novum Organum* mark the limits
of the present state of knowledge. In his view, the act of sailing past the columns towards *terrae incognitae* becomes a symbol of the advancement of knowledge that typified the Renaissance (Poirier, 2016), something to be sought after rather than feared:

'For why should a few received authors stand up like Hercules' columns, beyond which there should be no sailing or
discovering [...]?' (Bacon, 1973, p.61)

Our trusted guide Aristotle glances at the painting and sighs at the sight of the man who wrote about him with such scorn, before pointing to a corner of the office where scientists are holding a meeting. 'These souls,' says Aristotle, 'are searching

for ways of maximising happiness in science. Focusing on the consequences of scientists' actions, they believe that utility

should be the main criterion for good science,' he says. The Scientist listens in on the souls' conversation and realises she is

privy to a meeting of an equality, diversity and inclusion committee (Nature, 2018). Some scientists discuss their terms of

reference, while others propose enacting quotas to balance gender disparity in the sciences (Woolston, 2019). Eventually,

they decide it is necessary to gather further data, before moving forwards (Tzanakou, 2019). At a nearby table, souls read

papers and enter figures into calculators, while others weigh papers on electronic scales, before stacking them in neat piles.

'These souls,' says Aristotle, 'are evaluating funding proposals submitted by other scientists. Using complicated formulae,

they try to measure their usefulness to society.'

At a table close by, the Scientist and Aristotle see dozens of souls editing documents on large screens. 'Never satisfied, they

draft and redraft regulations and codes of conduct for laboratories, offices and conferences. In doing so, they serve at the

altar of deontology, believing that rules and regulations will rescue science,' says Aristotle grimly. The Scientist peers at

their screens. One of the souls is petitioning her university to sign up for policies that promote free access to research

publications (e.g. Else, 2019b), another is reading a list of recommendations around research assessment (e.g. San Francisco

Declaration on Research Assessment, 2012), while another is drafting regulations to protect the environment from the effects

of mining (e.g. Nature, 2019c).


Slowly, they walk towards an escalator leading to the top floor. 'As we rise, look around you: the souls on each floor are

closer and closer to living in accordance with the virtues,' says Aristotle, before pointing towards a meeting room with glass

walls around it. 'Over there, senior scientists are discussing how best to mentor students and develop a good working

atmosphere in their laboratory. The talk animatedly and trustingly of their own experience of science, and they share their

stories and their anxieties willingly with no thought of fear or penalty. In doing so, says Aristotle warmly, they are learning

to practice the virtues (e.g., Powell, 2018) : their efforts will bear fruit. Yet Aristotle also motions to a sadder sight: in offices

just down the corridor, scientists are working with policy officers, huddled over lists and protocols, as they try to develop

new ways of evaluating scientists. 'They know they must move away from metrics, to avoid the most egregious injustices'

he says, 'but, alas, they still rely on new regulations and HR formulations. And so they miss the vital point, that good science

comes from scientists' freedom to find their way, and develop their virtues.'

> *"It's a global reflection that we ask from our professors instead of quantified objectives [...]. It's not about listing all the outputs, it's about reflecting on a global perspective on their major achievements. It's much more qualitative than quantitative. [...]. We wanted to support people by giving them the chance to reflect on their own strengths and talents*
> 640    *and preferences, and what they wanted to do most in their career, and if they are good at it then it's fine for the university. [...] We very quickly came up with a completely revised regulatory framework, what we called our regulations for professorial staff, in which we as clearly as possible stipulated the framework in which the new evaluation method would be handled in the future."*

>     - University Policy Officer


At this point, Aristotle takes a seat by the window and invites the Scientist to sit down beside him. 'My friend, though well-intentioned, the attempts of these souls to create ethical science are an exercise in futility,' he says. 'Clearly, those who focus on the consequences of particular actions and weigh them up, trying to decide which will maximise happiness in science, are faced with an impractical and, ultimately, impossible calculation.' Aristotle pauses, before adding: 'And those who focus on

rules forget that rules are dependent on convention, and how can something dependent on convention be universally true?' 'Yes, I see,' says the Scientist.

'What the law prescribes, though just, is so only in an acquired or accidental way. It does not answer the vital question of how actions are to be performed and distributions to be made in order to be just' says Aristotle, reading from his *Ethics* (*Nic.*

V, 9). He points to a note on the margin:

*"Do we need a set of rules, a set of scientific commandments to actually behave properly? I don't think it does any harm. But I really do think that a lot of people know and have a strong sense of what's right, because ultimately that's what that's what we're searching for."*                                                                                          - PI


'The reality of science is a terrifying one, which codes of conduct and regulations can do little to assuage,' says Aristotle, before standing up. 'Furthermore, to focus on regulating the actions of scientists or on the consequences of those actions, is to miss the mark. Dear Scientist, the only way for science to thrive is to put the focus on scientists themselves as moral agents,' he says, pointing to a large scrawling on the wall:


ἦθος ἀνθρώπῳ δαίμων

(Auerbach, 1961)

(A man's character is his fate)

This phrase, known as the maxim of Heraclitus (Auerbach, 1961), is intended to remind the scientists in purgatory to redirect their focus from action to character. They stare at the scrawling for a time, until Aristotle breaks the silence: 'Walk with me, my friend'.

'Humans' purpose is to exercise their reason, and a good life is one in which reason is exercised well,' says Aristotle. 'Thus,
in order to be happy, which must be our ultimate goal in life, we must act in accordance with reason. And in the right and able exercise of reason, my friend, is where virtue resides,' he says. 'What do you mean by virtue, exactly?' asks the Scientist. 'Virtues are *dispositions* or settled habits of acting wisely which we praise in others,' (*Nic*. I, 13) replies Aristotle. 'Think of it this way: virtues are midpoints between extremes of excess and deficiency,' (Rachels, 2010) he adds. 'For example, courage is the midpoint between rashness and cowardice. To be good, we must choose the mean, avoiding both the
too much and the too little' (*Nic*. VI, 1).

'My friend, the virtues should be our guide in all matters, including science. Hence, as you look at the souls around you, remember this: although rules are useful, they must be guided by the virtues,' says Aristotle. 'Only by making a habit of cultivating the virtues, will a person flourish and fulfil his or her destiny,' he explains. 'I think I understand — scientists who practise the scientific virtues will necessarily do good science *and* be good scientists. Is that right?' the Scientist asks. Aristotle nods and adds: 'Moreover, the life of people who practice the virtues is inherently pleasant.'

Indeed, according to Aristotle, a person, in order to achieve εὐδαιμονία, the ancient Greek word that can be roughly translated as happiness, should cultivate the virtues. But it is worth noting that he made a distinction between intellectual virtues, which relate to the activity of reasoning, and are born from and cultivated through teaching, and moral virtues, which are not innate, but acquired by habit (Aristotle, 1953; Darwall, 2003; MacIntyre, 1998). Intellectual virtues include scientific knowledge, artistic or technical knowledge, intuitive reason, practical wisdom, and philosophic wisdom, while moral virtues include courage, temperance and justice (Darwall, 2003).

'Dear Scientist, the logic of science cannot save science, only the virtues have this power,' says Aristotle. 'Tell me, wise guide, which virtues are most appropriate for science?' asks the Scientist. Aristotle points to a marble staircase leading out into a sunny gallery.

## 6 *Paradiso* – a flourish of virtues

As we end our journey where Dante ended his, in heaven, we will glimpse at virtues that pertain to the life scientific (Fig. 1). The overwhelming brightness of the virtues makes the Scientist squint. Just as Virgil shielded Dante's eyes from the gaze of Medusa at the gates of Hell (*Inf*. IX, 58-60), Aristotle covers the Scientist's eyes with his hand, as they slowly adjust to their virtuous surroundings. What the Scientist starts to make out is a vision crafted by the imagination of modern-day scientists.

Some souls walk around aimlessly, while others tend to the plants in the courtyard. 'Here,' says Aristotle, 'the virtues are fully realised and signs of εὐδαιμονία are everywhere. Take note, for the words of present-day scientists are scripted upon the clouds':

*"I like to think about things. To my mind, that's the essence of science - not to spew it out."*  - PI

These words herald our first scientific virtue, reticence, in the sense of taking the time to reflect, and pausing before diving into scientific undertakings (Webster, 2019). Among scientists, Florentine polymath Leonardo da Vinci (1452-1519) was well-practiced in this virtue, given his 'habit of sometimes spending half a day at a time contemplating what he had done so

far,' (Vasari, 1965, p.12). Another is Charles Darwin (1809-1881), whose legendary delays in publication were occasioned by a desire to get things right, and find forms of expression best able to represent the complex paths of nature (Webster, 2016).

Next on our list of scientific virtues comes generosity, defined by Aristotle (*Nic*. IV, 1) as a mean between the extremes of

stinginess and extravagance (Rachels, 2010). In science, it applies to time, data, power and funding, commodities which need to be shared for good science to take place. This virtue is further elucidated in the words of scientists scrawled on the clouds:

*"You do need people who are willing to share and be generous with their ideas, and that's not just in terms of the*

*relationship between the lab manager and the postdocs or students, but among lab members themselves."*

- Professor

Our third scientific virtue is inspired by philosopher of science Paul Feyerabend (1924-1994), who famously argued that science should be free from the constraints of rules or 'method' (Feyerabend, 1993), 'anything goes' being 'the only

principle that does not inhibit progress' (1993, p.14). A fierce advocate of openness and diversity in science (Feyerabend,

1987), Feyerabend sought to dismantle the walls that keep science closed and stewing in its prejudice. 'There is no idea, however ancient and absurd, that is not capable of improving our knowledge,' he wrote (1993, p.33). Feyerabend championed a kind of innocent science, where non-scientists contribute to knowledge as much as scientists (e.g. Vera, 2018), and where secrecy and competition have no place. Innocence, then, in the sense of being open to others and to new ideas, is the name of this virtue (Webster, 2019), which incorporates trust and communicativity in its practice:

> *"The PI has to trust the people who are actually doing the experiments. [...] You have to trust that they will do them in a way that you have agreed, so if you say a particular dose or a particular time, then that's the experiment that they will do, or they will come back and say 'I think that I've made a mistake in the dose or I didn't have enough cells, or I've only got three time points instead of four', they will discuss that. They have to know that I'll treat them all fairly and [...] if I say I'm going to do something then I'll do it, and I won't be snapping at them or getting cross if they screw up, because that's just human."*
> — PI

> *"There's no solution [to co-authorship disputes] other than people should talk to one another from the beginning and throughout projects and make it clear: is this going to be co-authorship or not. [...] It's really [about] just communicating clearly what constitutes authorship and what doesn't."*
> — Technical Officer

Naturally, by practicing the virtue of innocence, scientists will open up to and take part in the pursuits of their wider communities. This is in line with Dante's admiration of civic life, which, he believed, necessitates active participation and communication from all community members (Honess, 2006). Through the virtue of innocence, scientists are able to listen attentively to lay people, gauge and benefit from their views, and find the right response to societal pressures. The virtuous clinical researcher consults their patients, and adjusts the research questions and the protocols accordingly.

The fourth scientific virtue is philia, or friendship, which was treated by Cicero (106-43BC) as inextricably linked to goodness:

'No one can be a friend unless he is a good man. But next to goodness itself, I entreat you to regard friendship as the finest

thing in all the world'

(Cicero, 1971, p.227).


Cicero wrote that 'it is from love, *amor*, that the word for friendship, *amicitia*, is derived' (1971, p.191). Thus, in being

loving and caring towards one another and towards non-human beings as well as to the planet, we exercise philia. In our

heavenly terrace of science, these words of modern-day scientists confirm philia to be an important ingredient to the good

life scientific:


*"You might just feel a bit down and having somebody in the lab can help you get over that, and that's really good. If*

*someone is having a hard time, I think that we need to recognise that and be kind to them and just think that they may*

*need a bit more help."*                                                                                                    - PI

On another cloud, the scientific virtue of humility is conjured up by these words:

*"I'm reasonably happy with the role I have right now - providing a lot of research support. [...] You do not just need the*

*shining professor stars; these people can only shine with a good support cast and I'm reasonably happy with my role in*

*this support cast now."*                                                                                    - Technical Officer


Humility in science, taken to stand in opposition to pride or hubris, can be understood in light of Aristotle's treatment of self-

love:

'It is right for the *good* man to be self-loving, because he will thereby himself be benefited by performing fine actions; and

by the same process he will be helpful to others. [...] For intelligence never fails to choose the course that is best for itself,

and the good man obeys his intelligence,' (*Nic*. IX, 8).

According to Aristotle, the 'good man' stands 'ready to sacrifice wealth, honours, all the prizes of life in his eagerness to

play a noble part,' (*Nic*. IX, 8). Humility, then, is properly dosed self-love, accompanied by a 'tempered desire for

excellence' (Boyd, 2014), standing in opposition to hubris and self-aggrandisement. For instance, when senior scientists

undergo training on how to mentor students and manage employees, they are learning to practice humility.

Honesty is the scientific virtue shining brightest among the clouds. It can be understood in the sense of truthfulness, both

towards others, as Aristotle defines it (*Nic*. IV, 7), and towards oneself. Aristotle's treatment of truthfulness applies to the

situations in which people present their accomplishments and commitments to others (*Nic*. IV, 7). This definition, which can

be applied to scholarly communication and publishing, does not, per se, provide a satisfactory account of scientific honesty,

for scientists must also direct truthfulness towards themselves, if they are to practice good science, as these words testify:

*"Honesty, [is the] number one [virtue]. It's a belief that, even though you may be absolutely convinced that you're right,*

*you may be wrong. It's very important to accept that the frame in which you're operating is not the total frame."*

- Emeritus Professor

Further guidance on this virtue comes from Dante himself, whose divine journey is one of self-consciousness (Shaw, 2014;

Wilson, 2011). In fact, only once the pilgrim has been honest with himself about his sins, can he reach God (*l'Amor che*

*move il sole e l'altre stelle*, 'The Love which moves the sun and the other stars', *Par*. XXXIII, 145).

'Dear Scientist, let us take heed,' says Aristotle. 'The six virtues we have contemplated are: reticence, generosity, innocence,

philia, humility, and honesty. These are the habits that a good, well-balanced scientist must practice and learn to embody.'

With tears in her eyes, the Scientist turns to her guide who says: 'Now that we have observed the conditions necessary for science to flourish, I must leave you here. I trust that you will treat what you have seen as a modern code of conduct for science'.

The Scientist wakes up with a start. Realising that her malaise had arisen from the state of the world around her, she knows
what to do. She must put the focus on human relationships. Her mission henceforth will be to restore the virtues by living according to them. Success in science will ensue. Keen to communicate what she has seen, she leaves the lab with a confident stride.

## 7 Conclusions


In this work, we have argued that virtue ethics provide an effective antidote for the ailments affecting modern science. Using the narrative structure of Dante's *Commedia*, we explored six scientific virtues, their corresponding vices, and various ethical approaches in science, some of which we consider to be misdirected. We have argued that, by practicing these virtues, and nurturing them as character traits, scientists will naturally be successful in their work. They will be 'good
scientists' in all senses. Specifically, we propose that neither utilitarian ethics (Mill, 2002) nor duty-based ethics (Rachels, 2010) alone, can rescue science. Moreover, rather than suggesting that utilitarian or duty-based ethics should complement virtue-based ethics (e.g. Resnik, 2012), we have made the case, through Aristotle's words, for the virtues to be the guides in all matters scientific, including the creation of rules.

Our interview quotes demonstrate that the scientists in our sample are deeply concerned with power dynamics and human relationships. On these aspects, codes of conduct and regulations say very little. Conversely, the narrative framework of Dante's *Commedia* and its moral universe, where humans have moral agency as individuals, provide a more personal, evocative and direct means of discussing ethical matters concerning character.

In addition to those identified during our interviews, further virtues remain to be explored. It is our hope that Dante's narrative and its characters, from Ulysses to Aristotle, will inspire future enquiries into matters of virtue ethics, should one wish to embark upon the journey.

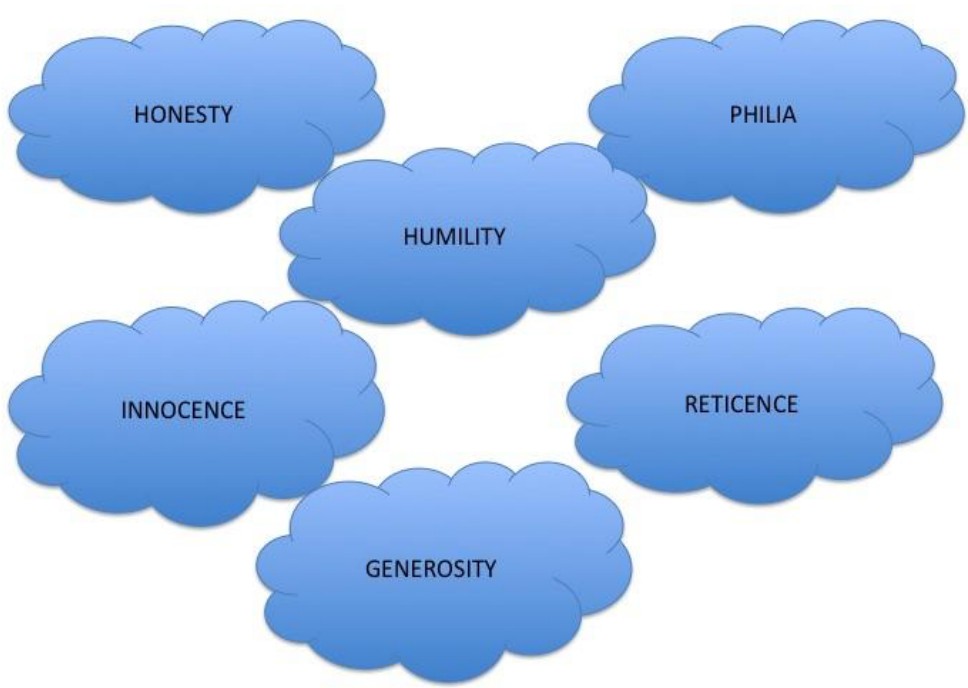


**Figure 1: The scientific virtues that emerged from interviews with scientists**.


**Appendix 1. Structure of the narrative**

| Inferno | | Purgatorio | Paradiso | |
|---|---|---|---|---|
| **Sins** | | | **Virtues** | |
| ↑<br><br><br><br><br>Severity increase and narrative direction<br><br><br><br>↑ | Dishonesty (fraud, corruption) | Misdirected love of science<br><br>(method of science: deontology and utilitarianism) | Honesty | ↑<br><br><br><br>Increase in brightness of the virtues and narrative direction<br><br><br><br>↑ |
| | Pride (hubris) | | Humility | |
| | Violence (including cruelty and power abuse) | | Philia | |
| | Fear, secrecy and competition | | Innocence (openness, communicativity, trust) | |
| | Greed (power, fame, money) | | Generosity | |
| | Unrestrained curiosity | | Reticence | |

**Appendix 2**

**Abbreviations**

*Commedia = Divina Commedia*

*Comedy = Divine Comedy*

*Inf. = Inferno*

*Purg. = Purgatorio*

*Par. = Paradiso*

E.g. (*Inf.* IV, 33-34, Singleton) = Fourth *canto* of Dante's *Inferno*, from the *Divine Comedy*; verses 33 to 34; English

translation by Singleton; full reference in bibliography).

*Met.* = Ovid's *Metamorphoses*

*Nic.* = Aristotle's *Nicomachean Ethics*

E.g. (*Nic.* V*,* 9) = *Nicomachean Ethics*, 5th book, 9th chapter

Translations are my own, unless otherwise stated.


**Data availability**

All data underlying the results is available in the manuscript. Additional data around this project is available from the

corresponding author. Transcripts of interviews are not provided to preserve anonymity of interviewees.


**Funding**

The authors received no financial sponsorship for this research.

## Competing Interests

The authors declare no competing interests.

## Author Contributions

AL and SW conceived and planned the study. AL conducted the interviews and drafted the manuscript, and SW and AL

worked on the manuscript.


## Acknowledgements

Ethical approval for the study was obtained by Imperial College London. We warmly thank all the people who spoke with us

about ethics in science and who inspired this work. This includes our interviewees and the Dublin Dante Summer School

participants and mentors. Warmest thanks go to our editor, Dr Rebecca Priestley, and to our referee, Dr Fabien Medvecky,

for their valuable feedback on this manuscript. We also wish to thank William Stapleton for his assistance with interview

transcription and Alexandra Fitzsimmons for her encouraging feedback on this work.

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
