# Peer review of "La Commedia Scientifica: Dante and the Scientific Virtues"

_Geoscience Communication, 2020_

## Referee Comment (RC1) · Fabien Medvecky (Referee) · 24 Jul 2020

Reversing Dante's tale, I'll start with paradise and the good. This is a beautiful written text, and narratively, I was happy to go along with it, almost no matter what it claimed. It was a lovely dance between ethics, scientific practice, and one of the great texts of classical literature. Scientifically, or, more precisely, academically, it was somewhat less robust. I'll divide my comments by purgatory (minor) and hell (major issues).

Purgatory: Also, not mentioning the mertonian norms of science seems a significant oversight. These set norms that are, at least potentially, akin to virtue, and some acknowledgement of their role needs to be present.

There are also some minor points. eg. Comparing science's ills with Florence's is

messy. Florence's ills were very contextualised. But here it confounds what is science's ills (reproducibility etc) and what are academia's ills. There is overlap, but these are not the same. So are the virtues virtues of scientists or academics? Indeed, the manuscript messily plays around the lack of definition of what counts as science, but this matters. In fact, we might think it matters when it comes to some of the virtues (does scientific humility require scientists to take non-scientific academically fields as equally relevant and worthwhile intellectual pursuits?) Hell: The paper opens with the tantalising promise to respond to the question: ""in contemporary science, is the good scientist also likely to be a successful one?" by looking at the moral world Dante presents in poem, specifically as work in virtue ethics. While the manuscript beautifully (I was going to say cleverly, but that would have missed the aesthetic virtues of the work) weaves a scientific parallel to Dante's guided journey during which a bunch of (scientist's) virtues emerge, the question as to whether this can make one a successful scientists (and the relationship between a 'good' scientist and a 'successful' scientist) is never returned to. Yet that's what I was sold on, so either the promise needs to change, or the delivery needs to amend to fulfil the promise. This is probably the major point.

The second issue, from a purely academic point, is what I can only think of as cherry picking (choosing one aspect or element to make a point and ignoring other potentially equally good aspects of elements), often without much justifying. For example, while Ulysses is potentially a good guide, it's not necessarily obvious, yet it flavours the work in a unique way. Similarly, in terms of which circles of hell are represented and mirrored for scientists (I'll limit myself to inferno) also shapes and somehow determines the discussion and interpretation. There is no discussion of the first circle, but issues around retrospectively problematic science - things/acts not thought at an issue at the time, but are now viewed an issue... eg p-hacking - seems like a good candidate (interestingly, Solzhenitsyn's gulag in the First Circle is a science and research gulag where scientists, due to the perceived value and virtue of science have a less-harsh gulag experience, which brings up interesting questions about the social placing of science). And likewise, there are no wood of the suicide. Maybe one of the sins/vices

of science is to sacrifice scientific success for the sake of other cares...at least this opens important discussions which the paper promised to respond to (see my first point)

I really enjoyed reading this, and I think it need to be out in the public sphere. It invites much needed reflection, and the invitation to draw on Dante's journey is apt. If the intellectual rigour can be made to match the prose, this would make one hell of a paper.

---

## Referee Comment (RC2) · Anonymous Referee #2 · 17 Sep 2020

I have just finished reading the paper. I was intrigued by the title (DANTE AND THE SCIENTIFIC VALUES) but I was disappointed by the actual paper. I thought it would be about the relationship between ethical virtues and scientific research, but in fact it is about the moral character (i.e. the behavior) of researchers. The fact that the researchers considered in the paper are scientists seems irrelevant, they could be humanists or social scientists, business people or carpenters and the thrust of the paper would still be the same and equally applicable. The idea of using the Divine Comedy as a grid to evaluate the behavior (the moral standards) of contemporary researchers is brilliant but in the end it does not tell us anything new about them beyond the fact that they are subject to human failings; above all, it doesn't tell us anything about scientific values, which I mistakenly thought would be the subject of the paper.

---

## Author Comment (AC1) · 14 Dec 2020

**Referee 1**

*Referee comment*

Reversing Dante's tale, I'll start with paradise and the good. This is a beautiful written text, and narratively, I was happy to go along with it, almost no matter what it claimed. It was a lovely dance between ethics, scientific practice, and one of the great texts of classical literature. Scientifically, or, more precisely, academically, it was somewhat less robust. I'll divide my comments by purgatory (minor) and hell (major issues).

*Author reply*

We wish to thank Dr Medvecky for a very helpful and thoughtful review. We reply to each of the comments below. Our suggested edits in the paper are in blue below, with line numbers indicating where we wish to make the changes.

*Referee comment*

Purgatory: Also, not mentioning the mertonian norms of science seems a significant oversight. These set norms that are, at least potentially, akin to virtue, and some acknowledgement of their role needs to be present.

*Author reply*

We agree that the Mertonian norms should feature in our exploration of virtue ethics in science, given their key importance in the sociology of science. Whatever their plausibility, these norms sit firmly in the tradition that searches for rules and codes as guarantors of the safe-keeping of science. Yet their over-simplification and idealisation of science is significant to any study of scientific virtue, and to capture this debate we will refer also to Ian Mitroff's (1974) opposing counter-norms, developed from his three and a half-year study of the Apollo lunar mission scientists. Mitroff's argument against Merton was that, morally-speaking, actual scientists are highly plastic ('ambivalent') in their ethical judgements, being both aware of the norms, and aware also of the justifications for ignoring them.

Specifically, we would like to include Merton's norms in our Purgatory of science, where modern scientists are striving to be virtuous, but their love of science is misdirected (much like love of God was misdirected in Dante's Purgatorio), or, as Mitroff suggests, it faces in two directions. Their love of science is manifested both by their selfless and disinterested behavior, and also by their selfish and obsessive strivings. To convey this torment of the contemporary scientist, we propose adding a section to the room where scientists are trying to save science through allegiance to a scientific method (p. 21, line 509):

'Here, reside the souls of scientists whose love of science is misdirected, for they are trying to save science by sticking like limpets to a simple view of a simple scientific method,' says Aristotle, pointing towards a group of scientists is walking around in circles, reciting Merton's norms (Merton, 1973) under their breath, ad infinitum, a kind of liturgy. The Scientist

recognises those utterings as the norms, or behaviours, described by American sociologist Robert Merton (1910-2003) when during WWII he sought 'institutional imperatives' that crucial to the ethos of science (Merton, 1973). These are: universalism, the notion that scientific claims should be evaluated irrespective of social or personal factors; communism, the idea that scientific findings belong to, and should be shared with, society; disinterestedness, the idea that scientists are not motivated by self-interest; and organized skepticism, the notion that scientists must subject their ideas to careful scrutiny. Aristotle, that great observer of how things are, cautions us however against the trap of over-simplifying human behavior just to make us feel better: 'If only humans could be so pure' he declares, and then: 'Nowhere in science can so clear a gaze be found!'. And the far-sighted Greek points to Ian Mitroff's (1974) portrayal of capricious, ambitious scientists working on the Apollo lunar missions, all of them 'looking both ways'. Mitroff, confronted by the truly complex paths of contemporary scientists, and their abandonment of the 'one way to truth', duly formulated, for each of Merton's norms, an opposing version, a liturgical heresy, the so-called 'counter-norms'. He argued that both norms and counter-norms existed in science, and had to.  For instance, he writes (Mitroff, 1974, p. 579), 'if universalism is rooted in the impersonal nature of science, an opposing counter-norm is rooted in the personal character of science.' He made central then to the scientific effort an unstable and oscillatory ambivalence. Naturally, for scientists unwilling to reflect honestly on their lives, and facing hard battles for grants and esteem, the time-consuming and apparently inefficient dynamics of professional ambivalence, becomes a source of trouble and anguish.

'These scientists desperately seek a moral compass,' says Aristotle. For each norm they so wistfully whisper, the words of the scientists interviewed by Mitroff materialise on a giant whiteboard behind them, and taunt them: 'Science is an intensely personal enterprise, it knows no simple rules' the Scientist reads. Aristotle looks gently at our struggling hero and speaks: 'These scientists seek simple truths, but forget the fraught ways of all professional culture. In running from that truth, they cast themselves into a place where nothing is as it seems, and where no door leads to the contentment of a flourishing career.

'Have you learned well? Yes? Then let us proceed, dear Scientist' says Aristotle, and gestures towards a table where a group of scientists work together.

**Referee comment**

There are also some minor points. eg. Comparing science's ills with Florence's is messy. Florence's ills were very contextualised. But here it confounds what is science's ills (reproducibility etc) and what are academia's ills. There is overlap, but these are not the same. So are the virtues virtues of scientists or academics? Indeed, the manuscript messily plays around the lack of definition of what counts as science, but this matters. In fact, we might think it matters when it comes to some of the virtues (does scientific humility require scientists to take non-scientific academically fields as equally relevant and worthwhile intellectual pursuits?)

**Author reply**

We agree that it is important to note that many of the issues that emerge from our ethical analysis, as well as virtues and vices, are valid for academia and research as a whole, not just for science. Arguably however, critical discussion of the virtues and vices of science, compared to those of the humanities, have received especial scrutiny in the last two decades, because of its great expansion (a key element of this is the turbo-charging of the life sciences through genetic techniques), and because of the way this expansion links to a growing culture of competitiveness, speed, and 'publish or perish'. This explains our emphasis on science here, both as a culture and as a multidisciplinary pursuit including natural sciences, social sciences, and physical sciences, but we see now that a clarification is necessary in the manuscript.

We would like to add the following lines to the Methodology (line 166):

'We use the term science to denote the natural, physical, and social sciences, but we wish to point out that many of the virtues and vices identified in this paper are pertinent to academic research and issues of research integrity as a whole. Our choice to restrict our investigation to science, which because of its high research costs and societal impacts is in truth a zone of particular academic pressure, is reflected in our choice of interviewees (see below).'

We see how the Florence metaphor (Florence as science) needs some further elaboration and should perhaps be tempered to a simile. We think this parallel is a good one because of the cruelty and arbitrary nature of Florentine politics, which saw good people exiled, and was marked by a kind of short-sightedness, with endless changes in legislation and regulations, love of immediate wealth, and inter-familial rivalries. Florence is a valid reference for our investigation of science because we remember it also for its contribution to the making of the Renaissance, and therefore the rational and scientific Europe, through the work of Galileo in astronomy, Brunelleschi in architecture and the development of realistic perspective, and Machiavelli in statecraft. We think adding more detail on this, including Dante's own invective against Florence, will help make our case stronger.

We therefore would like to change Line 160 "science *is* that sick woman", to "science resembles that sick woman".

We would also like to add the following to further illustrate our parallel between Florence and science, and have it replace the current lines 141-148:

Dante's work cannot be understood without understanding the political turmoil of Florence in the poet's time. Indeed, Dante's complicated relationship with his native Florence permeates the *Commedia* (Hainsworth & Robey, 2015). The Florentines and their feuds are particularly well represented in the *Inferno*, as these ironic verses attest to:

Godi, Fiorenza, poi che se' sì grande
che per mare e per terra batti l'ali,
e per lo 'nferno tuo nome si spande!
(*Inf.* xxvi, 1-3)

(Be joyous, Florence, you are great indeed,
for over sea and land you beat your wings;

through every part of Hell your name extends!)

In Dante's lifetime, Florence was one of many city-states in northern and central Italy (Scott, 2004). A wealthy mercantile and banking centre, it was deeply troubled by factional struggles and changes in legislation and systems of government (Shaw, 2014). At the turn of the fourteenth century, Florence was a city divided, with nobles and wealthy merchants splitting into Guelfs and Ghibellines, respectively the party of the Pope, and that of the Holy Roman Emperor, although petty rivalries tended to override ideological affiliations (Shaw, 2014). The inhabitants of Florence did not just have internal feuds to contend with, but also spats with other Italian cities. As pointed out by writer Andrew Norman Wilson in his book *Dante in Love* (2011):

'The inhabitants of medieval Italian cities lived in a state of such enmity with one another that it was necessary for them to live huddled in fortified towers.'

The Ghibellines came to power in Florence in 1260, following the battle of Montaperti, but were in turn expelled from the city in 1266, after the Guelf victory at the Battle of Benevento (Santagata, 2016; Shaw 2014). In 1295, the Guelf party in Florence divided into two factions, known as Blacks and Whites (Santagata, 2016), a split that would lead to Dante's traumatic exile. Dante was an ambitious man, actively involved in public life. By 1300, Dante's party, the Guelfs, had been in power for more than thirty years (Shaw, 2014). From June to July of 1300, Dante, who aligned with the White Guelfs, served as one of six priors who exercised executive power in Florence. In 1301, while he was absent from the city, the Blacks staged a coup and returned to power, partly owing to an intervention by Pope Boniface VIII (Wilson, 2011). The following year, Dante was fined, charged with corruption, and exiled. It is in exile that he started work on the *Commedia* (around 1308), a task that would consume him until shortly before his death (Shaw, 2014).

**Referee comment**

Hell: The paper opens with the tantalising promise to respond to the question: "" in contemporary science, is the good scientist also likely to be a successful one?" by looking at the moral world Dante presents in poem, specifically as work in virtue ethics. While the manuscript beautifully (I was going to say cleverly, but that would have missed the aesthetic virtues of the work) weaves a scientific parallel to Dante's guided journey during which a bunch of (scientist's) virtues emerge, the question as to whether this can make one a successful scientists (and the relationship between a 'good' scientist and a 'successful' scientist) is never returned to. Yet that's what I was sold on, so either the promise needs to change, or the delivery needs to amend to fulfil the promise. This is probably the major point.

**Author reply**

We completely agree that this question is a central one in this manuscript. We allude to it and hint at the answer early on in the paper, in lines 37-38, when Aristotle is introduced. We return to it in Purgatory, when Aristotle says that scientists who practice the scientific virtues will necessarily do good science *and* be good scientists, lines 601-603. Indeed, the

notion that, by practicing the virtues, people will flourish in whatever their chosen exercise, is one of Aristotle's key teaching points in his *Ethics.* So we can be confident that our answer to this question is in line with Aristotle's virtue ethics, applied to science. The point of our paper is to suggest that the great struggle Dante captures and indeed recommends in his epic poem, a struggle Dante sees as a route to understanding and fulfilment as a human, has parallels in the dilemmas and challenges the contemporary scientist also faces. While these challenges and injustices are far from un-remarked in the academic press, the solutions normally recommended centre on management devices; our paper, while respectful of these policies, puts emphasis instead on the ethical principle of 'the well-lived scientific life', and proposes Aristotle and Dante as sources of guidance, both for individuals and institutions.

Prompted by the referee's comment, we would like to further emphasise this at the end, and suggest adding the following sentence to our conclusions, line 734:

"...we explore six scientific virtues, their corresponding vices, and various ethical approaches in science, some of which we consider to be misdirected. We have argued that, by practicing these virtues, and nurturing them as character traits, scientists will naturally be successful in their work. They will be 'good scientists' in all senses"

We would also like make the following edit to the very end of the narrative, line 727, to further solidity this point:

'Her mission henceforth will be to restore the virtues by living according to them. Success in science will ensue. Keen to communicate what she has seen, she leaves the lab with a confident stride."

**Referee comment**

The second issue, from a purely academic point, is what I can only think of as cherry picking (choosing one aspect or element to make a point and ignoring other potentially equally good aspects of elements), often without much justifying. For example, while Ulysses is potentially a good guide, it's not necessarily obvious, yet it flavours the work in a unique way.

**Author reply**

We see the need for further justification for our choice of trusted guide. We have chosen Ulysses as a guide for many reasons, not least because of Dante's admiration of his curiosity. Indeed, Dante's fascination with Ulysses' intellect shines through the *Commedia*, and, of course, curiosity is often equated with science. So Ulysses typifies the dangers of curiosity taken too far, not tempered by the virtues. But there are other reasons why Ulysses makes a good guide too. Ulysses is often portrayed as a wise man and an accomplished orator, but he is also a somewhat mixed character: vengeful, boastful, neglectful, as well as cunning and brave. Many Dante scholars argue that, in the *Commedia* (canto XXVI)*,* he effectively tricks his crew into taking sail, knowingly leading them to a watery grave. Yet, Ulysses evokes fierce admiration in others (as in Dante the character, who yearns to talk to

him in Hell). Thus the figure of Ulysses, so capable of evoking opposite reactions in different people, is first and foremost an ambivalent character, and it is precisely this ambivalence that make him a very good guide for scientists. Indeed, as we say earlier, the ambivalence of scientists has been recognised as indeed central to science, as laid out by the important contributions of Merton (1942, 1973) and Mitroff (1974), who describe the way conflicting norms in the institution of science generate marked ambivalence in the lives of scientists, as well as the sure-footed development of robust scientific theory. Our modern guide, one imagines, will have seen all sides of life, and be able to relate very well to the struggles we encounter in Hell.

But, when we get to page 20, Ulysses having done his work, we must hand over to Aristotle, an acknowledged moral force, a teacher rather than a warrior, and a philosopher more capable of showing us the ethical highlands. Dante chose well, and indeed Aristotle's philosophy permeates *The Divine Comedy.*

We would like to add more explanation and signal that the reader will first come across Ulysses as Guide no 1, and then Aristotle as Guide no 2. We suggest deleting the words 'such as Aristotle" from line 137, and adding the following after line 139:

"For now, suffice it to say that we will encounter three main characters: a Scientist, and two guides, first Ulysses and later Aristotle. In our choice of characters, we weave together the thoughts and the questions of the Scientist, Ulysses and Aristotle, and Dante the character, whose journey we follow, Virgil, who represents reason, and, finally, Beatrice, who represents divine knowledge. But before we set off, let us consider Dante's world and how it can speak to modern science."

We would also like to further explain our choice of Ulysses by highlighting his ambivalence, when we first encounter him at the gates of Hell (line 265):

"Clearly Ulysses is somewhat of a mixed character: vengeful, boastful, neglectful, as well as cunning and brave. This ambivalence is also a trait that has long been identified as pertaining to scientists by sociological studies of scientists (Mitroff, 1974). Indeed, by Mitroff's account, the ability to hold diametrically opposing views in mind at the same time, may be essential to science."

**Referee comment**

Similarly, in terms of which circles of hell are represented and mirrored for scientists (I'limit myself to inferno) also shapes and somehow determines the discussion and interpretation. There is no discussion of the first circle, but issues around retrospectively problematic science - things/acts not thought at an issue at the time, but are now viewed an issue. . . eg p-hacking - seems like a good candidate (interestingly, Solzhenitsyn's gulag in the First Circle is a science and research gulag where scientists, due to the perceived value and virtue of science have a less-harsh gulag experience, which brings up interesting questions about the social placing of science). And likewise, there are no wood of the suicide. Maybe one of the sins/vices of science is to sacrifice scientific success for the sake of other cares. .

.at least this opens important discussions which the paper promised to respond to (see my first point).

*Author reply*

We thank the referee for highlighting these structural differences and for prompting us to investigate Solzhenitsyn's fascinating work. In our paper, for the sake of brevity, we are not able to include all the circles in the *Commedia* (this would make for a much longer epic!). We have included a Dark Wood to kick off the Scientist's journey, but, chiefly, we have chosen to follow the Aristotelian-Ciceronian subdivision of incontinence-violence-fraud (Scott, 2004, p. 196) that is found in Dante's Hell (see lines 287-291). So, in our scientific *Commedia*, we start with the sins of incontinence (unrestrained curiosity and greed for fame, power and money), before moving onto the sins of violence, directed towards oneself or other people, animals or our planet, (secrecy, competition and fear, cruelty and power abuse), and culminating with pride and dishonesty (fraud, corruption).

Having identified these vices and placed them within our Hell, we derive from them our six virtues described in Paradiso. Our Purgatorio is typified, as in Dante, by misdirected love. Indeed this is our focus. While in Dante, it is love of God that is misdirected, in our case, it is love of science. Hence, in our paper, we use Dante's themes to illuminate the ways modern science fails, and the ways we can attempt to rescue it. In short, our work is edifying and works by selective, rather than comprehensive, use of Dante. This selectivity is shown by the way we do not even come close to the detailed subdivisions encountered on Dante's mountain of Purgatorio, with its seven terraces. Our aim is to stimulate the reader into new ways of looking at the ethics of science research culture.

At the beginning of the Inferno (line 288), we write:
"Before proceeding, it is worth noting that the sins we will encounter here and in our scientific Purgatorio only broadly follow those of Dante's moral universe. In common with the Inferno, we will consider incontinence, violence and fraud (Auerbach, 1961; Di Zenzo, 1965), but certain deviations from the Commedia's nine circles and seven terraces of purgatory must be allowed, if pertinence to science is to be our guiding principle." Overall we are sure our principle of using Dante's great poem as a metaphor for the discontents of science has every merit, and will be amply justified as the paper develops its argument

We would also like to delete the word "certain" from the above lines, and replace it with "substantial" to make clear that our structure will differ quite a lot.

Finally, we would like to make another small addition to our upper reaches of Purgatory. We intend this to further illustrate that, as we ascend our Purgatorio, modern scientists are getting closer and closer to the virtues. These lines will add more hope to the narrative (after line 561):

''Slowly, they walk towards an escalator leading to the top floor. 'As we rise, look around you: the souls on each floor are closer and closer to living in accordance with the virtues,' says Aristotle, before pointing towards a meeting room with glass walls around it. 'Over

there, senior scientists are discussing how best to mentor students and develop a good working atmosphere in their laboratory. The talk animatedly and trustingly of their own experience of science, and they share their stories and their anxieties willingly with no thought of fear or penalty. In doing so, says Aristotle warmly, they are learning to practice the virtues: their efforts will bear fruit. Yet Aristotle also motions to a sadder sight: close by offices just down the corridor, scientists are working with policy officers, huddled over lists and protocols, as they try to develop new ways of evaluating scientists. 'They know they must move away from metrics, to avoid the most egregious injustices' he says, 'but, alas, they still rely on new regulations and HR formulations. And so they miss the vital point, that good science comes from scientists' freedom to find their way, and develop their virtues.'

"It's a global reflection that we ask from our professors instead of quantified objectives [...]. It's not about listing all the outputs, it's about reflecting on a global perspective on their major achievements. It's much more qualitative than quantitative. [...]. We wanted to support people by giving them the chance to reflect on their own strengths and talents and preferences, and what they wanted to do most in their career, and if they are good at it then it's fine for the university. [...] We very quickly came up with a completely revised regulatory framework, what we called our regulations for professorial staff, in which we as clearly as possible stipulated the framework in which the new evaluation method would be handled in the future."

- University Policy Officer

**Referee comment**

I really enjoyed reading this, and I think it needs to be out in the public sphere. It invites much needed reflection, and the invitation to draw on Dante's journey is apt. If the intellectual rigour can be made to match the prose, this would make one hell of a paper.

**Author reply**

We warmly thank Dr Medvecky for his review.

**References**

Merton, R.K. 1973. Sociology of science: theoretical and empirical investigations. Chicago: Chicago University Press.

Mitroff, I. 1974. Norms and counter-norms in a select group of the Apollo moon scientists: a case study of the ambivalence of scientists. American Sociological Review, 39(4), 579-595.

---

## Author Comment (AC2) · 14 Dec 2020

**Referee 2**

*Referee comment*

I have just finished reading the paper. I was intrigued by the title (DANTE AND THE SCIENTIFIC VALUES) but I was disappointed by the actual paper. I thought it would be about the relationship between ethical virtues and scientific research, but in fact it is about the moral character (i.e. the behavior) of researchers. The fact that the researchers considered in the paper are scientists seems irrelevant, they could be humanists or social scientists, business people or carpenters and the thrust of the paper would still be the same and equally applicable. The idea of using the Divine Comedy as a grid to evaluate the behavior (the moral standards) of contemporary researchers is brilliant but in the end it does not tell us anything new about them beyond the fact that they are subject to human failings; above all, it doesn't tell us anything about scientific values, which I mistakenly thought would be the subject of the paper.

*Author reply*

We thank the referee, but politely disagree that the paper does not shed any new light on the moral character of scientists. The notion that the human failings highlighted in our manuscript could apply to other groups of people beyond scientists may well be true, but this does not detract from our analysis here, which was based on in-depth interviews with scientists about the life scientific. All quotes and findings in the narrative are pertinent to our sample of scientists, and from them, using a qualitative thematic analysis that is in keeping with grounded theory, we attempt to derive a set of scientific vices and virtues that can guide scientists, and act as a moral compass through contemporary science. Furthermore, we argue, the ethical issues explored in the manuscript, from the pace of scientific publishing, to the problems connected with research integrity and research assessment, are all deeply embedded in scientific culture and will be familiar to any scientist reading our paper. We note that the referee has misread the title of the manuscript, which is: "La Commedia Scientifica: Dante and the Scientific Virtues", not "Dante and the scientific values". A study of scientific values is beyond the scope of what we have done here.

---

## Author Response (AR1)

Dear Dr Priestley,

Thank you very much for your feedback on our manuscript. We have made the changes suggested in our response to Reviewer 1. We have also addressed the proofing notes (see below). We wish to thank you for a very pleasant and constructive editorial and review process.

Best wishes,

Dr Anthea Lacchia
* * *
Line 54: add the word "be" after "likely to"
This has been added.

Lines 174-175: the statement "we believe we have achieved diversity" seems a bit vague. What is the basis for this belief? Given that you didn't "formally gather demographic information"? Can you give a bit more information here, to be more precise and clear about what you are saying?
Yes, we see how this was vague and have rephrased to: 'Despite a relatively small sample size, our sample included a range of ages and career stages, and a gender ratio of 7 females to 6 males, although we did not formally gather demographic information.' (Lines 243-244)

Line 204: Add a para break
This has been added.

Line 739: Given the small sample size, it is too much of a stretch to claim "quotes demonstrate that scientists are deeply concerned with ... "?
That's a good point. We have rephrased to: 'Our interview quotes demonstrate that the scientists in our sample are deeply concerned with power dynamics and human relationships.' (Lines 825-826)